#  Impact of l-Carnitine Supplementation on Liver Enzyme Normalization in Patients with Chronic Liver Disease: A Meta-Analysis of Randomized Trials

**DOI:** 10.3390/jpm12071053

**Published:** 2022-06-27

**Authors:** Hyunwoo Oh, Chan Hyuk Park, Dae Won Jun

**Affiliations:** 1Department of Internal Medicine, Uijeongbu Eulji Medical Center, Eulji University School of Medicine, Uijeongbu 11690, Korea; asklepios1258@eulji.ac.kr; 2Department of Internal Medicine, Hanyang University Guri Hospital, Hanyang University College of Medicine, Guri 11923, Korea; 3Department of Internal Medicine, Hanyang University College of Medicine, Seoul 04763, Korea

**Keywords:** carnitine, acetylcarnitine, carnitine-orotate, chronic liver disease

## Abstract

The effectiveness of l-carnitine in chronic liver disease remains controversial. We conducted this meta-analysis to assess the efficacy of various forms of l-carnitine in the treatment of chronic liver disease. Methods: We searched the Cochrane Library, EMBASE, KMBASE, and Medline databases for all relevant studies published until April 2022 that examined the ability of l-carnitine or its derivatives to normalize liver enzymes in patients with chronic liver disease. We performed meta-analyses of the proportion of patients with alanine aminotransferase (ALT) normalization and post-treatment serum aspartate aminotransferase (AST) and ALT levels. A random effects model was used for meta-analyses. Results: Fourteen randomized controlled trials (1217 patients) were included in this meta-analysis. The proportion of patients in whom ALT normalized was higher in the carnitine-orotate treatment group than in the control group (pooled odds ratio (OR), 95% confidence interval (CI) = 4.61 (1.48–14.39)). The proportion of patients in whom ALT normalized was also higher among those who received the carnitine-orotate complex, a combination of carnitine-orotate, biphenyl dimethyl dicarboxylate, and other minor supplementary compounds than in those who did not without significant heterogeneity (pooled OR (95% CI) = 18.88 (7.70–46.27); df = 1; *p* = 0.51; I^2^ = 0%). l-carnitine supplementation effectively lowered serum ALT levels compared to controls (pooled mean difference (95% CI) = −11.99 (−22.48 to −1.49)). Conclusions: l-carnitine supplementation significantly lowered ALT and AST levels and normalized ALT levels in patients with chronic liver disease.

## 1. Introduction

l-carnitine (3-hydroxy-4-*N*-trimethylammonium butyrate) is an endogenous compound that plays a pivotal role in fatty acid metabolism [1]. l-carnitine supplementation is marketed in the form of acetyl-l-carnitine (analogs of l-carnitine) [2] and carnitine-orotate (l-carnitine and an ion complex salt of orotic acid) [3,4]. Commercially, the carnitine-orotate complex, a combination of carnitine-orotate, biphenyl dimethyl dicarboxylate, and other minor supplementary compounds, is also available [5].

l-carnitine is mainly derived from foods, such as meat and dairy products. It is also synthesized in the human liver and kidneys; in other words, l-carnitine deficiency occurs more frequently in patients with chronic liver disease or liver cirrhosis [2,6]. Carnitine acts as a carrier for fatty acids across the mitochondrial membrane for subsequent β-oxidation and reduces hepatic free fatty acids, causing decreased triglyceride accumulation in the cytoplasm of hepatocytes [7]. Moreover, it plays an important role in reducing oxidative stress and increasing proinflammatory cytokine expression [8]. l-carnitine, acetyl-l-carnitine, and carnitine-orotate supplements have been used as adjuvant therapies in chronic liver disease, including non-alcoholic steatohepatitis, alcoholic fatty liver disease, viral hepatitis, and cirrhosis [9,10,11].

Accordingly, many studies have investigated the effect of l-carnitine on liver enzymes and liver disease, but the results are inconclusive. In a previous meta-analysis, a pooled analysis suggested that l-carnitine supplementation significantly decreased serum liver enzyme levels [12,13]. However, the previous meta-analysis included healthy volunteers as well as patients with chronic liver disease. In addition, previous studies analyzed only l-carnitine, and many randomized controlled trials (RCTs) on various forms of carnitine were omitted. Therefore, this meta-analysis investigated the effects of l-carnitine supplementation, including acetyl-l-carnitine and carnitine-orotate, in patients with chronic liver disease.

## 2. Materials and Methods

### 2.1. Search Strategy

We searched the Cochrane Library, EMBASE, KMBASE and Medline databases for all relevant studies published between January 1990 and April 2022 that examined the ability of carnitine to normalize liver enzymes in patients with chronic liver disease. The following search string was used: ((carnitine) OR (l-carnitine) OR (carnitine-orotate) OR (carnitine-orotate) OR (carnitine complex) OR (godex) OR (hepadif) OR (biphenyl dimethyl dicarboxylate) OR (BDD) OR (DDB)) AND ((liver) OR (hepatic) OR (hepatitis) OR (cirrhosis)). The detailed search strategies used for each database are presented below.

Cochrane library

#1: carnitine or L-carnitine or ‘carnitine-orotate’ or ‘carnitine orotate’ or ‘carnitine complex’ or godex or hepadif or ‘biphenyl dimethyl dicarboxylate’ or BDD or DDB;

#2: liver or hepatic or hepatitis or cirrhosis;

#3: #1 and #2 (with publication year from 1990 to 2022, in Trials).

EMBASE (search interface: Ovid)

1: ((carnitine or L-carnitine or carnitine-orotate or carnitine orotate or carnitine complex or godex or hepadif or biphenyl dimethyl dicarboxylate or BDD or DDB) and (liver or hepatic or hepatitis or cirrhosis)).ab,ti;

2: Limit 1 to (embase and yr=“1990–Current”). 

KMBASE

(((((((((([ALL=carnitine] OR [ALL=L-carnitine]) OR [ALL=carnitine-orotate]) OR [ALL=carnitine orotate]) OR [ALL=carnitine complex]) OR [ALL=godex]) OR [ALL=hepadif]) OR [ALL=biphenyl dimethyl dicarboxylate]) OR [ALL=BDD]) OR [ALL=DBB]) AND ((([ALL=liver] OR [ALL=hepatic]) OR [ALL=hepatitis]) OR [ALL=cirrhosis])).

MEDLINE (Search interface: PubMed)

(carnitine[tw] OR L-carnitine[tw] OR (carnitine-orotate[tw]) OR (carnitine orotate[tw]) OR (carnitine complex[tw]) OR godex[tw] OR hepadif[tw] OR (biphenyl dimethyl dicarboxylate[tw]) OR BDD[tw] OR DDB[tw]) AND (liver[tw] OR hepatic[tw] OR hepatitis[tw] OR cirrhosis[tw]) AND (“1990/01/01”[Date—Publication]: “3000”[Date—Publication]).

### 2.2. Inclusion/Exclusion Criteria

The inclusion criteria were as follows: (a) patients: diagnosis of chronic liver disease; (b) intervention: taking the target drugs, including carnitine-orotate, carnitine-orotate complex, and l-carnitine; (c) comparator: no target drugs; (d) outcome: proportion of patients with alanine aminotransferase (ALT) normalization or difference in post-treatment serum aspartate aminotransferase (AST) and ALT levels between the treatment and control groups; and (e) study design: RCT (Table 1). Non-human studies or abstract-only publications were excluded.

### 2.3. Study Selection

Duplicates from multiple search engines were removed from the literature search results. After the titles and abstracts were reviewed, irrelevant studies were excluded based on the inclusion and exclusion criteria. Thereafter, we reviewed the full text of all the remaining studies. Two investigators (H.O. and C.H.P.) independently assessed the eligibility of each study. Any disagreements were resolved by discussion and consensus. If no agreement was reached, a third investigator (D.W.J.) determined the final eligibility.

### 2.4. Quality Assessment

Two investigators (H.O. and C.H.P.) independently assessed the methodological quality of the individual studies using the Cochrane Risk of Bias assessment tool for RCTs [14]. 

### 2.5. Data Extraction

The data were extracted using a form developed in advance. Two investigators (H.O. and C.H.P.) independently extracted the following information: first author, year of publication, study design, study period, country, baseline characteristics of the study participants, dosage of target drugs, proportion of participants with ALT normalization, post-treatment serum AST and ALT levels, and other major findings of each study. If values for the meta-analysis were not sufficiently reported by the individual studies, we asked the corresponding author for the data.

### 2.6. Study Endpoints

The primary endpoint of this meta-analysis was the proportion of patients with post-treatment ALT normalization. The secondary endpoint was the mean difference in the post-treatment serum AST and ALT levels between the treatment and control groups. 

### 2.7. Statistical Analysis

As most studies of carnitine-orotate (or the carnitine-orotate complex) investigated the proportion of patients with ALT normalization, pooled odds ratios (ORs) with 95% confidence intervals (CIs) were calculated. A random-effects model was used. For a meta-analysis of studies that reported continuous variable outcomes (e.g., post-treatment serum AST and ALT levels), mean differences (MDs) and 95% CIs between the treatment and control groups were calculated. 

We assessed heterogeneity using two methods: Cochran’s Q test, wherein *p* values < 0.1 were considered statistically significant for heterogeneity, and I^2^ statistics, wherein values > 50% were suggestive of significant heterogeneity [15]. Based on Cochrane recommendations, publication bias was assessed when ≥10 studies were included [14]. Publication bias was assessed using a funnel plot [16], a radial plot [17], Egger’s test [18], and the AS-Thompson test [19]. We performed sensitivity analyses after excluding studies (1) involving decompensated cirrhosis; and (2) with a high risk of bias. Additionally, we performed a jackknife sensitivity analysis after excluding one different study each time.

All P values were two-tailed, and those <0.05, except on the heterogeneity test, were considered statistically significant. Analyses and reporting were performed in accordance with the Preferred Reporting Items for Systematic Reviews and Meta-Analyses guidelines [20]. All statistical analyses were conducted using Review Manager 5.3 (version 5.3.5; Cochrane Collaboration, Copenhagen, Denmark) and R (version 4.4.4; R Foundation for Statistical Computing, Vienna, Austria).

Details of the protocol for this systematic review were registered on PROSPERO and can be accessed at https://www.crd.york.ac.uk/prospero/display_record.php?ID=CRD42022332856 (accessed on 28 May 2022)[21].

## 3. Results

### 3.1. Study Selection and Characteristics

Fourteen studies including 1217 patients were included in this meta-analysis (Figure 1). The characteristics of the included studies are summarized in Table 2. The studies were published during 2000–2016 and included enrollment periods during 2000–2014 [22,23,24,25,26,27,28,29,30,31,32,33,34,35]. Among the 14 studies included, four investigated the efficacy of carnitine-orotate or carnitine-orotate complex and reported the proportion of patients with post-treatment ALT normalization [22,23,24,25]. The other 10 studies assessed the impact of l-carnitine on serum AST and ALT levels [26,27,28,29,30,31,32,33,34,35]. Of the 14 included studies, one involved patients with decompensated cirrhosis [31]. 

The risk of bias assessment is shown in Appendix A. Five (36%) patients had an unclear risk of bias in the domain of random sequence generation because random sequence generation methods were not clearly described in those studies. The other nine studies (64%) had a low risk of bias for random sequence generation. The risk of allocation concealment was unclear in 13 (93%) studies. All studies were assessed as having a low risk of performance and detection bias because the current study outcomes were less likely to be affected by participant and investigator blinding. Incomplete outcome data and selective reporting were not identified in any RCT. Four studies (27%) were assessed as having a high risk of other biases because baseline AST or ALT levels differed between the treatment and control groups despite randomization [26,31,32,34]. 

### 3.2. Impact of Carnitine-Orotate (or Carnitine-Orotate Complex) on ALT Normalization

The impact of carnitine-orotate (or carnitine-orotate complex) on ALT normalization is shown in Figure 2. In studies on carnitine-orotate, the proportion of patients with ALT normalization was higher in the treatment group versus that in the control group (pooled OR (95% CI) = 4.61 (1.48–14.39)). Heterogeneity was not identified (degrees of freedom (df) = 1, *p* = 0.16, I^2^ = 49%). In studies of the carnitine-orotate complex, the proportion of patients with ALT normalization was also higher in the treatment versus control group without significant heterogeneity (pooled OR (95% CI) = 18.88 (7.70–46.27); df = 1, *p* = 0.51, I^2^ = 0%). 

Since two studies on carnitine-orotate analyzed the efficacy of different drug dosages (carnitine-orotate 900 mg vs. 600 mg), we further performed dosage subgroup analyses (Appendix A). When the usual dosage for clinical purposes (carnitine-orotate 900 mg) was used, it effectively normalized ALT levels, as described in the main results (Figure 2). However, the efficacy of lower-dosage carnitine-orotate (600 mg) was not identified in the subgroup analysis (pooled OR (95% CI) = 1.27 (0.66–2.44); df = 1, *p* = 0.72, I^2^ = 0%). 

### 3.3. Impact of l-Carnitine (or Acetyl-l-Carnitine) Supplementation on Serum AST and ALT Levels

Since 10 studies on l-carnitine or acetyl-l-carnitine supplementation assessed and reported the main outcomes as continuous variables (i.e., serum AST and ALT levels), the mean difference in post-treatment AST or ALT levels between the treatment and control groups was analyzed (Figure 3). In a meta-analysis of post-treatment AST levels, l-carnitine or acetyl-l-carnitine supplementation effectively lowered serum AST levels compared to the control (pooled MD (95% CI) = −15.84 (−24.56 to −7.13)). The post-treatment serum ALT level was lower in the treatment group than in the control group (pooled MD (95% CI) = −11.99 (−22.48 to −1.49)). Significant heterogeneity was identified in both meta-analyses (AST level: df = 9, *p* < 0.01, I^2^ = 94%; ALT level: df = 9, *p* < 0.01, I^2^ = 96%). Egger’s test and the AS-Thompson test for publication bias found no significant asymmetry in the funnel plot and radial plot (*p* > 0.1) (Appendix A).

### 3.4. Sensitivity Analysis

Two sensitivity analyses were performed. First, we performed a sensitivity analysis of l-carnitine or acetyl-l-carnitine supplementation on serum AST and ALT levels after excluding a study of patients with decompensated cirrhosis [31] (Appendix A). Post-treatment AST levels were also lower in the l-carnitine or acetyl-l-carnitine supplementation group than in the control group (pooled MD (95% CI) = −17.39 (26.90 to −7.88)). The post-treatment ALT levels tended to be lower in the treatment group than in the control group despite no significant difference (pooled MD (95% CI) = −10.25 (−21.60 to 1.10)). Second, in the sensitivity analysis, after the exclusion of four studies showing different baseline characteristics in AST or ALT levels [26,31,32,34] (Appendix A), post-treatment AST and ALT levels were also lower in the l-carnitine or acetyl-l-carnitine supplementation group than in the control group (pooled MD (95% CI): AST level, −25.77 (−32.92 to −18.62); ALT level, −17.14 (–27.89 to −6.38)). In jackknife sensitivity analyses (Appendix A), a significant study that affected the pooled meta-analysis results was not identified. These sensitivity analyses showed the robustness of our meta-analyses. 

## 4. Discussion

The current meta-analysis found that carnitine-orotate and carnitine-orotate complexes taken at the usual dosage significantly affected ALT normalization in patients with chronic liver disease. We also showed that l-carnitine (or acetyl-l-carnitine) supplementation significantly decreased serum AST and ALT levels. Our findings suggest that l-carnitine and its derivatives have beneficial effects in normalizing liver enzymes in patients with chronic liver disease. The novelty of this study lies in the fact that we extracted and analyzed data from studies of patients with chronic liver disease and elevated liver enzyme levels. Previous meta-analyses suggested the efficacy of l-carnitine supplementation for reducing serum liver enzyme levels, similar to our study findings; however, some individual studies in the previous meta-analyses involved healthy volunteers or patients with liver enzyme elevations caused by reasons other than chronic liver disease [12,13]. In addition, clinically useful information can be provided through the analysis of carnitine-orotate complex containing other minor supplements that have not been previously analyzed.

AST and ALT are enzymes that transport the amino groups of aspartate and alanine to ketoglutaric acid. AST is present in the liver and other organs, including the muscle, while ALT is primarily present in the liver; thus, ALT is a more specific marker of hepatocellular injury than AST [36]. We adopted the odds ratio of the ALT normalization rate as an important analysis target. In real-world practice, clinicians aim to normalize ALT in patients with various chronic liver diseases to ensure a satisfactory prognosis [37,38,39,40,41]. In recent studies, elevated liver enzymes were associated with significantly increased liver-related mortality, and normal-range AST and ALT levels were associated with a lower risk of death [42,43]. Therefore, l-carnitine and its derivatives may help improve the prognosis of patients with chronic liver disease and elevated liver enzyme levels. 

There are considerable limitations to the present study. First, there was substantial heterogeneity in the included studies, although the current meta-analysis had a limited study population of patients with chronic liver disease. Interstudy heterogeneity was observed due to various disease etiologies, including chronic viral hepatitis B, chronic viral hepatitis C, and non-alcoholic fatty liver disease, as well as different dosages and durations of l-carnitine supplementation. Therefore, further studies are needed to examine the time- and dose-dependent effects of l-carnitine supplementation. Second, it is also important to note that our study findings should not be generalized to patients worldwide since studies on carnitine-orotate were conducted only in Korea and studies on l-carnitine or acetyl-l-carnitine were conducted only in Italy and other countries. In Italy, the majority of studies were conducted by the same research group. In the future, it will be necessary to assess the efficacy of l-carnitine supplementation in various regions and races worldwide. Third, we should be careful when interpreting the effect size since medications other than l-carnitine (e.g., antiviral agents) were administered to the treatment and control groups in 6 of the 14 enrolled studies. Finally, the patient follow-up period of the studies included in this paper is relatively short, within one year. As mentioned above, the relationship between the improvement of ALT and the prognosis of the disease is emphasized, so long-term follow-up studies are needed in the future.

## 5. Conclusions

In conclusion, the pooled data from RCTs provide evidence of the beneficial effects of l-carnitine supplementation for lowering serum ALT and AST levels. Our findings also showed that carnitine-orotate can be effective in patients with chronic liver diseases with intervention doses of 900 mg/day for ALT normalization. 

## Figures and Tables

**Figure 1 jpm-12-01053-f001:**
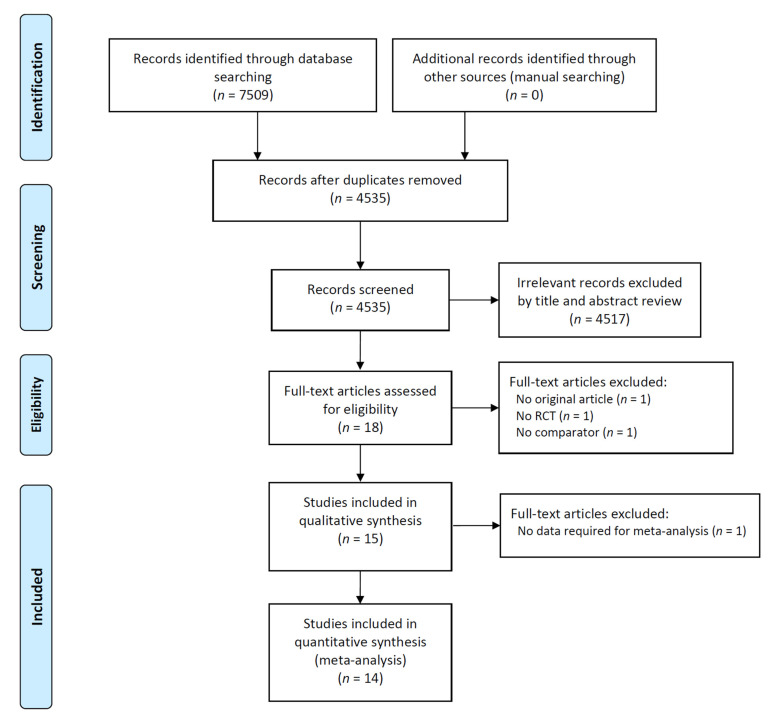
Study flow diagram.

**Figure 2 jpm-12-01053-f002:**
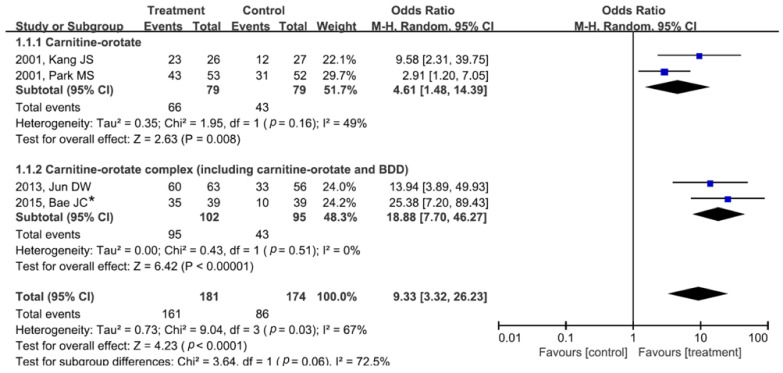
Forest plot of alanine aminotransferase normalization after carnitine-orotate (or carnitine-orotate complex) versus control treatment. The carnitine–orotate complex includes carnitine-orotate, BDD, and other minor supplementary compounds. The dosage of carnitine-orotate in the included studies in this analysis was 900 mg per day, which is the usual dosage for clinical purposes. * The values in this study were provided by the corresponding author. BDD, biphenyl dimethyl dicarboxylate; M-H, Mantel–Haenszel; CI, confidence interval; df, degrees of freedom.

**Figure 3 jpm-12-01053-f003:**
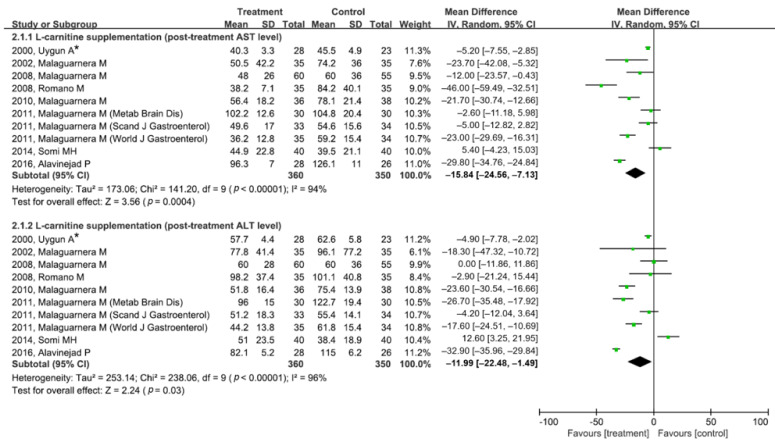
Forest plot of post-treatment serum AST and ALT levels after l-carnitine (or acetyl-l-carnitine) supplementation versus control treatment. Parentheses following the first author’s name indicate the journal in which the study was published. * Multiple doses were investigated in this study. Data from the maximal dosage group (3 g/day) were used in the analysis. ALT, alanine aminotransferase; AST, aspartate aminotransferase; CI, confidence interval; df, degrees of freedom; IV, inverse variance; SD, standard deviation.

**Table 1 jpm-12-01053-t001:** Inclusion criteria of the present meta-analysis. ALT, alanine aminotransferase; AST, aspartate aminotransferase.

Inclusion Criteria	Description
Population	Patients with chronic liver disease
Intervention	Taking the target drugs, including carnitine-orotate, carnitine-orotate complex, and L-carnitine
Comparison	No target drugs
Outcomes	Proportion of patients with ALT normalization or difference in post-treatment serum AST and ALT levels
Study design	Randomized controlled trial

**Table 2 jpm-12-01053-t002:** Baseline characteristics of the included studies.

Publication Year	First Author	Study Design	Study Period	Country	Study Population	Inclusion of Patients with Decompensated Cirrhosis	Treatment	Control	Other Medication (Both Treatment and Control Groups)	Number of Participants	Age, Years, Mean ± SD	Male Sex, %	BMI, kg/m^2^, Mean ± SD	Baseline Laboratory Findings		Risk of Bias
														AST, IU/mL,Mean ± SD	ALT, IU/mL,Mean ± SD	
2001	Kang JS [22]	Double-blinded RCT	2000	Korea	Chronic hepatitis	No	Treatment A: Carnitine-orotate (900 mg/day) Treatment B: Carnitine-orotate (600 mg/day)	Placebo	BDD 150 mg/day	95	Treatment A: 49.0 ± 9.7; Treatment B: 41.6 ± 11.7; Control: 44.0 ± 11.6	Treatment A: 63.6; Treatment B: 63.3; Control: 71.9	N/A	N/A	N/A	Unclear
2001	Park MS [23]	Double-blinded RCT	N/A	Korea	Chronic hepatitis	No	Treatment A: Carnitine-orotate (900 mg/day) Treatment B: Carnitine-orotate (600 mg/day)	Placebo	BDD 150 mg/day	154	Treatment A: 44.6 ± 11.5; Treatment B: 43.2 ± 13.0; Control: 45.6 ± 13.8	Treatment A: 92.5; Treatment B: 77.6; Control: 82.7	N/A	NA	N/A	Unclear
2013	Jun DW [24]	Open-label RCT	N/A	Korea	Chronic hepatitis B	No	Carnitine-orotate complex (900 mg/day as carnitine-orotate)	No carnitine-orotate complex	Entecavir 0.5 mg/day	130	Treatment: 43.0 ± 9.8; Control: 44.9 ± 10.0	Treatment: 63.5 Control: 66.1	N/A	Treatment: 118.8 ± 70.3; Control: 111.3 ± 70.2	Treatment: 159.4 ± 67.5; Control: 160.3 ± 74.9	Unclear
2015	Bae JC [25]	Double-blinded RCT	2011–2012	Korea	NAFLD with type 2 DM	No	Carnitine-orotate complex (900 mg/day as carnitine-orotate)	Placebo		78	Treatment: 50.6 ± 9.3; Control: 52.0 ± 9.4	Treatment: 64.1; Control: 74.4	Treatment: 28.2 ± 2.6; Control: 26.7 ± 3.7	Treatment: 61.8 ± 25.5; Control: 51.7 ±22.1	Treatment: 94.9 ± 36.4; Control: 79.2 ± 27.2	Low
2000	Uygun A [26]	Open-label RCT	N/A	Turkey	NAFLD	N/A	Treatment A: l-carnitine 1 g/day; Treatment B: 2 g/day; Treatment C: 3 g/day	No l-carnitine		101	Treatment A: 43.4 ± 10.8; Treatment B: 42.5 ± 9.7; Treatment C: 40.0 ± 9.1; Control: 42.6 ± 7.3	Treatment A: 79.2; Treatment B: 61.5; Treatment C: 75.0; Control: 69.6	Treatment A: 4.4 ± 2.3; Treatment B: 26.2 ± 3.1; Treatment C: 25.4 ± 2.7; Control: 25.8 ± 2.5	Treatment A: 57.5 ± 3.9; Treatment B: 47.8 ± 2.1; Treatment C: 51.0 ± 6.3; Control: 46.3 ± 6.0	Treatment A: 72.3 ± 5.4; Treatment B: 71.5 ± 4.6; Treatment C: 75.1 ± 7.0; Control: 70.5 ± 7.3	High
2002	Malaguarnera M [27]	Open-label RCT	N/A	Italy	Chronic hepatitis C	No	L-carnitine 2 g/day	No L-carnitine	IFN-α 3 million IU three times a week	70	Treatment: 56.8 ± 7.2; Control: 57.7 ± 6.1	Treatment: 62.9; Control: 60.0	Treatment: 26 ± 1.9; Control: 26 ± 2.4	Treatment: 110 ± 86; Control: 114 ± 79	Treatment: 186 ± 99; Control: 163 ± 108	Unclear
2008	Malaguarnera M [28]	Double-blinded RCT	2000–2003	Italy	Minimal hepatic encephalopathy	No	Acetyl-L-carnitine 4 g/day	Placebo		115	Treatment: 48 ± 10; Control: 45 ± 11	Treatment: 55.0; Control: 63.6	Treatment: 24.8 ± 3.1; Control: 25.1 ± 3.3	Treatment: 68 ± 31; Control: 67 ± 32	Treatment: 71 ± 40; Control: 68 ± 44	Unclear
2008	Romano M [29]	Open-label RCT	2000–2003	Italy	Chronic hepatitis C	No	L-carnitine 2 g/day	No -carnitine	IFN-α 3 million IU three times a week + ribavirin 1000 mg	70	Treatment: 50.1 ± 6.1; Control: 50.4 ± 5.6	Treatment: 56.7; Control: 53.3	Treatment: 25.8 ± 3.1; Control: 25.7 ± 3.2	Treatment: 125.0 ± 46.2; Control: 116.0 ± 49.3	Treatment: 162.0 ± 49.2; Control: 156.0 ± 47.4	Unclear
2010	Malaguarnera M [30]	Double-blinded RCT	2004–2006	Italy	NASH	N/A	L-carnitine 2 g/day	Placebo		74	Treatment: 47.9 ± 5.4; Control: 47.8 ± 5.8	Treatment: 55.6; Control: 52.6	Treatment: 26.6 ± 3.7; Control: 26.5 ± 3.8	Treatment: 132.8 ± 14.7; Control: 135.4 ± 15.1	Treatment: 125.7 ± 12.9; Control: 120.2 ± 12.8	Unclear
2011	Malaguarnera M(Metab Brain Dis)^a^ [31]	Double-blinded RCT	2002–2006	Italy	Severe hepatic encephalopathy	Yes	Acetyl-L-carnitine 4 g/day	Placebo		60	Treatment: range, 37–64; Control: range, 35–65	Treatment: 46.7; Control: 50.0	N/A	Treatment: 119.2 ± 13.1; Control: 114.2 ± 24.5	Treatment: 106.7 ± 15.7 ^b^; Control: 136.3 ± 31.0	High
2011	Malaguarnera M(Scand J Gastroenterol)^a^ [32]	Double-blinded RCT	2002–2005	Italy	Minimal hepatic encephalopathy	No	Acetyl-L-carnitine 4 g/day	Placebo		67	Treatment: range, 37–65; Control: range, 34–67	Treatment: 60.6; Control: 55.9	N/A	Treatment: 102.1 ± 15.2 ^b^; Control: 80.4 ± 19.8	Treatment: 117.4 ± 16.0 ^b^; Control: 90.2 ± 14.3	Unclear
2011	Malaguarnera M(World J Gastroenterol)^a^ [33]	Open-label RCT	2004–2007	Italy	Chronic hepatitis C	No	L-carnitine 4 g/day	Placebo	Peg-IFN-α 2b 1.5 μg/kg/week + rivavirin 800–1200 mg/day (adjusted to body weight)	69	Treatment: 47.6 ± 4.9; Control: 47.1 ± 5.4	Treatment: 62.9; Control: 58.8	Treatment: 27.1 ± 3.1; Control: 27.4 ± 2.9	Treatment: 145.0 ± 44.2; Control: 136.0 ± 41.4	Treatment: 182.1 ± 46.2; Control: 174.1 ± 42.2	Unclear
2014	Somi MH [34]	Open-label RCT	2012–2014	Iran	NAFLD	No	L-carnitine 1 g/day	No L-carnitine		80	Treatment: 40.3 ± 7.8; Control: 41.1 ± 8.3	Treament: 82.5; Control: 82.5	Treatment: 29.4 ± 3.9; Control: 28.6 ± 3.2	Treatment: 60.5 ± 28.3; Control: 52.6 ± 24.4	Treatment: 81.7 ± 40.1 ^b^; Control: 54.1 ± 17.6	High
2016	Alavinejad P [35]	Double-blinded RCT	N/A	Iran	NAFLD with type 2 DM	No	L-carnitine 2.25 g/day	Placebo		54	Treatment: 60 ± 5; Control: 59 ± 9	Treatment: 75.0; Control: 65.4	Treatment: 28.6 ± 4.6; Control: 29.5 ± 3.6	Treatment: 122.7 ± 13.6; Control: 125.3 ± 14.0	Treatment: 124.0 ± 11.3; Control: 120.0 ± 10.8	Unclear

^a^ Parentheses indicate the journal in which the study was published. ^b^ There was a significant difference in baseline values between treatment and control groups. ALT, alanine aminotransferase; AST, aspartate aminotransferase; BDD, biphenyl dimethyl dicarboxylate; BMI, body mass index; DM, diabetes mellitus; DNA, deoxyribonucleic acid; IFN, interferon; IU, international unit; N/A, not available; NAFLD, non-alcoholic fatty liver disease; NASH, non-alcoholic steatohepatitis; RCT, randomized controlled trial; SD, standard deviation.

## Data Availability

All relevant data are included in the study and supplementary information.

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
