# Peer review of "Impact of l-Carnitine Supplementation on Liver Enzyme Normalization in Patients with Chronic Liver Disease: A Meta-Analysis of Randomized Trials"

_jpm, 2022, doi:10.3390/jpm12071053_

Round 1

Reviewer 1 Report

In this manuscript, Dr. Oh and colleagues evaluated the clinical impact of L-carnitine Supplementation on liver enzymes in patients with chronic liver diseases using systematic review and meta-analysis methods. The method and study process was adequate and written well. However, I have some concerns before acceptance. Specific comments are given below.

1.           This study showed that 900mg/day, not 600mg/day, carnitine-orotate can be useful for ALT normalization. However, as described as a limitation, this meta-analysis and systematic review were mainly based on Korea(n=5), and Italy(n=7). Can the authors perform the sub-analysis by region (Asia vs West)? In addition, please assess the effects of L-carnitine by etiology.

2.           How were the influences of the other nutrients and supplementations (e.g. BCAA, zinc) excluded?

3.           The authors described that it is unnecessary to assess the publication bias when <10 studies were included based on the Cochrane recommendations. However, the funnel plot and Egger’s test may be needed for assessing publication bias since this study includes 15 studies. 

Author Response

Reviewer 1 
> Thank you for your good comment, and thank you again for your work as a reviewer.

1. This study showed that 900mg/day, not 600mg/day, carnitine-orotate can be useful for ALT normalization. However, as described as a limitation, this meta-analysis and systematic review were mainly based on Korea(n=5), and Italy(n=7). Can the authors perform the sub-analysis by region (Asia vs West)? In addition, please assess the effects of L-carnitine by etiology.
> Thank you for the good point. As suggested in Figure 2 and Figure 3, pooled analysis of ORs was conducted only in Korean studies and pooled analysis of mean difference was conducted in studies of Italy (7), Turkey (1), and Iran (2). Since the analysis method and target are different, it is difficult to analyze the comparison between East Asia and the West. In the case of sub-analysis by etiology, minimal or severe hepatic encephalopathy is the result of several etiology. Sub-analysis is difficult to perform because the cause of progression to hepatic encephalopathy is mixed in each study.

2. How were the influences of the other nutrients and supplementations (e.g. BCAA, zinc) excluded?
> Thank you for the good point. The availability of other supplementations is specified in the exclusion criteria of the individual studies included. In the presented analysis, the pooled analysis of ORs was performed for each carnitine-orotate and carnitine-orotate complex containing other minor supplements to exclude the effect of other minor supplements.

3. The authors described that it is unnecessary to assess the publication bias when <10 studies were included based on the Cochrane recommendations. However, the funnel plot and Egger’s test may be needed for assessing publication bias since this study includes 15 studies.
> Thank you for the good point. We evaluated publication bias using the Egger’s test for 10 studies that included in the analysis of pooled MD. AS-Thompson test was also performed in consideration of heterogeneity. 

[Please see the attachment. Funnel plot, Radial plot in word file]

# Egger's test p-value 0.5642
# AS-Thompson test p-value 0.675
This result means that no significant asymmetry was found in both analyses. We revised [Method-2.7 Statistical analysis] as follows: “Based on Cochrane recommendations, publication bias was assessed when ≥10 studies were included. Publication bias was assessed using a funnel plot, Egger’s test, and AS-Thompson test.” And we presented the result of Egger’s test and AS Thompson test at the end of [3.3 Impact of L-carnitine (or acetyl-L-carnitine) supplementation on serum AST and ALT levels] as follows and added supplement figure 3: “In Egger’s test and AS-Thompson test for publication bias found no significant asymmetry in the funnel plot (p > 0.1) (Figure S3).”

Reviewer 2 Report

dear colleagues,

thank you for this submission.

title is better be informative than query. something like - in patients with chronic liver disease, l-carnitine administration dramatically reduced alt and ast levels: findings of a systematic review and meta-analysis

abstract – re order databases alphabetically + if space permit add details of the statistical pooling techniques

keywords are random and repeat of the title please use mesh to generate better keywords to make your systematic review and meta-analysis (srma) visible https://meshb-prev.nlm.nih.gov/

introduction was adequate however did not justify srma lines 57-58 no systematic review or meta-analysis has examined its effects on chronic liver disease alone is not really a good justifications authors should expand on the novelty of this study please see prisma2020 updated guidelines

protocol registration should be at start

search strategy was complete and thanks for adding the search syntax however please move it to main text no need for appendix it made reading difficult.

please add pico statement as a table perhaps before search strategy

srma endpoints not justified clinically why authors selected these + malaguarnera 2011 published two studies in 2011 are you sure its not same patients? check with (d.w.j.) the third author who resolved disagreements and report this in results explicitly.

data analysis was very basic and did not perform additional needed work for example within studies heterogeneity not reported only between using i2 we need to see within so please calculate tau and tau2

perform jackknife sensitivity analysis to see if certain studies have influenced results i have concerns about studies uygun 2000, malaguarnera 2011/ after reading the results i realized some sensitivity analysis was done but was not reported in methods please take both of my comments at once here

i would like to see eggers regression test for publication bias not only funnel or if you want add radial plot so we see effect size distribution/bias

figure 1 is confusing it says 14 studies included in final analysis buy abstract and text report 15 including 1,269 patients figure 1 is poor quality it need to be redrawn using better techniques i always use r software for statistical techniques

table 1 can be better improved for readability its perfect for information but very poor for presentation

add one additional column to state overall risk of bias also

risk of bias vert clear i like figure s1 but this is not rob summary this is traffic light plot please add summary plot discuss this with expert statistician/meta-researcher

discussion is very brief i will expand on the strengths and weaknesses. i will also add a section for clinicians about the importance of this srma also give recommendations for future studies

general comments/suggestions

please don’t refer to your manuscript as study it’s a review specifically its srma

j of personal medicine/mdpi is an open access so better to add high quality images in main text/manuscript and avoid supplu because its hustle to move between files to read and view visuals

Author Response

Reviewer 2

> Thank you for your good comment, and thank you again for your work as a reviewer.

1) title is better be informative than query. something like - in patients with chronic liver disease, l-carnitine administration dramatically reduced alt and ast levels: findings of a systematic review and meta-analysis

> Thank you for the good point. We revised it as follows: "Impact of L-Carnitine Supplementation on Liver Enzymes Normalization in Patients with Chronic Liver Disease: A Meta-Analysis of Randomized Trials."

2) abstract – re order databases alphabetically + if space permit add details of the statistical pooling techniques

> Thank you for the good point. We revised the methods of abstract as follows:

“Methods: We searched the Cochrane Library, EMBASE, KMBASE, and Medline databases for all relevant studies published until April 2022 that examined the ability of L-carnitine or its derivatives to normalize liver enzymes in patients with chronic liver disease. We performed meta-analyses of the proportion of patients with alanine aminotransferase (ALT) normalization and post-treatment serum aspartate aminotransferase (AST) and ALT levels. A random-effects model was used for meta-analyses.”

3) keywords are random and repeat of the title please use mesh to generate better keywords to make your systematic review and meta-analysis (srma) visible https://meshb-prev.nlm.nih.gov/

> Thank you for the good point. We corrected “L-carnitine” to “Carnitine” and “acetyl-L-carnitine” to “Acetylcarnitine”. There were no results for carnitine-orotate and chronic liver disease in Main Heading Terms.

4) introduction was adequate however did not justify srma lines 57-58 no systematic review or meta-analysis has examined its effects on chronic liver disease alone is not really a good justifications authors should expand on the novelty of this study please see prisma2020 updated guidelines

> Thank you for the good point. We have revised it as follows: ”However, the previous meta-analysis included healthy volunteers as well as patients with chronic liver disease. In addition, previous studies analyzed only L-carnitine, and many randomized controlled trials (RCTs) on various forms of carnitine were omitted

5) protocol registration should be at start

> Protocol registration was conducted on 2022-05-17, and a meta-analysis was conducted after protocol registration.

6) search strategy was complete and thanks for adding the search syntax however please move it to main text no need for appendix it made reading difficult.

> We moved the appendix to below 2.1 Search strategy and reordered it alphabetically.

7) please add pico statement as a table perhaps before search strategy

> Thank you for the good point. We added Table 1 below.

Inclusion criteria

Description

Population

Patients with chronic liver disease

Intervention

Taking the target drugs, including carnitine-orotate, carnitine-orotate complex, and L-carnitine

Comparison

No target drugs

Outcomes

Proportion of patients with ALT normalization or difference in post-treatment serum AST and ALT levels

Study design

Randomized controlled trial

 Table 1. Inclusion criteria of the present meta-analysis. ALT, alanine aminotransferase; AST, aspartate aminotransferase

8) srma endpoints not justified clinically why authors selected these + malaguarnera 2011 published two studies in 2011 are you sure its not same patients? check with (d.w.j.) the third author who resolved disagreements and report this in results explicitly.

> Thank you for the good point. (Malaguarnera Scand J Gastroenterol 2011) investigated minimal hepatic encephalopathy patients, and (Malaguarnera Metab Brain Dis 2011) investigated severe hepatic encephalopathy (grade 3 of the West Haven). In other words, those two studies included different study participants each other. Therefore, we included both studies in our meta-analysis.

9) data analysis was very basic and did not perform additional needed work for example within studies heterogeneity not reported only between using i2 we need to see within so please calculate tau and tau2

> Thank you for your comment. We presented the result of tau2 in the main figures and supplement figures.

10) perform jackknife sensitivity analysis to see if certain studies have influenced results i have concerns about studies uygun 2000, malaguarnera 2011/ after reading the results i realized some sensitivity analysis was done but was not reported in methods please take both of my comments at once here

> Thank you for your comment. We presented the result of jackknife sensitivity analyses at the end of [3.4 Sensitivity analysis] as follows and added supplement figure 6: “In jackknife sensitivity analyses (Figure S6), a significant study which affected the pooled meta-analysis results was not identified. These sensitivity analyses showed the robustness of our meta-analyses.”

[Please see the attachment. jackknife sensitivity analyses in word file]

11) i would like to see eggers regression test for publication bias not only funnel or if you want add radial plot so we see effect size distribution/bias

> Thank you for your comment. We evaluated publication bias using the Egger’s test for 10 studies that were included in the analysis of MD. AS-Thompson test was also performed in consideration of heterogeneity.

[Please see the attachment. Funnel plot, Radial plot in word file]

# Egger's test p-value 0.5642

# AS-Thompson test p-value 0.675

This result means that no significant asymmetry was found in both analyses. We revised [Method-2.7 Statistical analysis] as follows: “Based on Cochrane recommendations, publication bias was assessed when ≥10 studies were included. Publication bias was assessed using a funnel plot, Egger’s test, and AS-Thompson test.” And we presented the result of Egger’s test and AS Thompson test at the end of [3.3 Impact of L-carnitine (or acetyl-L-carnitine) supplementation on serum AST and ALT levels] as follows and added supplement figure 3: “In Egger’s test and AS-Thompson test for publication bias found no significant asymmetry in the funnel plot (p > 0.1) (Figure S3).”

12) figure 1 is confusing it says 14 studies included in final analysis buy abstract and text report 15 including 1,269 patients figure 1 is poor quality it need to be redrawn using better techniques i always use r software for statistical techniques

> Thank you for the good point. We will exclude study [ref 20 (Hong ES)], which was included only in the qualitative evaluation, for fear that readers will also be confused. Accordingly, the contents of the table and the manuscript have been revised. Also, I will increase the quality of figure 1 by attaching the pdf file.

13) table 1 can be better improved for readability its perfect for information but very poor for presentation

> Thank you for the good point. Now we revised Table 2 (original Table 1) to improve the readability as your recommendation.

14) add one additional column to state overall risk of bias also

> We agree with your comment. Overall risk of bias was added in Figure S1 and Table 2 with one additional column.

[Please see the attachment. Overall risk of bias in word file]

15) risk of bias vert clear i like figure s1 but this is not rob summary this is traffic light plot please add summary plot discuss this with expert statistician/meta-researcher

> Thank you for the good point. We revised the title of Figure S1 as follows: “Traffic light plot for risk of bias in individual studies.” We also clarified the criteria of assessing overall risk of bias as follows: “*For the overall risk of bias, low risk of bias required random sequencing, allocation concealment, and blinding in order to be scored low risk with no other important concerns; unclear risk of bias was assigned if 1 or 2 domains were scored not clear or not done; high risk of bias was assigned if >2 domains were scored not clear or not done.”

16) discussion is very brief i will expand on the strengths and weaknesses. i will also add a section for clinicians about the importance of this srma also give recommendations for future studies

> Thank you for the good point. Strengths and weaknesses were added as follows, respectively: ”In addition, clinically useful information can be provided through analysis of carnitine-orotate complex containing other minor supplements that have not been previously analyzed.“, "Finally, the patient follow-up period of the studies included in this paper is relatively short, within one year. As mentioned above, the relationship between the improvement of ALT and prognosis of the disease is emphasized, so long-term follow-up studies are needed in the future."

17) please don’t refer to your manuscript as study it’s a review specifically its srma

> Thank you for the good point. Now we specified our study as a meta-analysis through the manuscript.

Round 2

Reviewer 1 Report

The authors modified this manuscript correctly based on the reviewers' suggestions. 

Reviewer 2 Report

dear authors, thank you very much for addressing my concerns. 

one more request is to provide a very quick analysis and results of trim and fill because according to funnel and radial plots + statistical results that there is some publication bias.